# Node-Loss Detection Methods for CZ Silicon Single Crystal Based on Multimodal Data Fusion

**DOI:** 10.3390/s23135855

**Published:** 2023-06-24

**Authors:** Lei Jiang, Rui Xue, Ding Liu

**Affiliations:** 1School of Automation and Information Engineering, Xi’an University of Technology, Xi’an 710048, China; jianglei@xaut.edu.cn (L.J.); xueruui@163.com (R.X.); 2Crystal Growth Equipment and System Integration National & Local Joint Engineering Research Center, Xi’an University of Technology, Xi’an 710048, China

**Keywords:** CZ silicon single crystal, node-loss detection, continuous wavelet transform, convolutional neural network, attention mechanism, multimodal data fusion

## Abstract

Monocrystalline silicon is an important raw material in the semiconductor and photovoltaic industries. In the Czochralski (CZ) method of growing monocrystalline silicon, various factors may cause node loss and lead to the failure of crystal growth. Currently, there is no efficient method to detect the node loss of monocrystalline silicon at industrial sites. Therefore, this paper proposed a monocrystalline silicon node-loss detection method based on multimodal data fusion. The aim was to explore a new data-driven approach for the study of monocrystalline silicon growth. This article first collected the diameter, temperature, and pulling speed signals as well as two-dimensional images of the meniscus. Later, the continuous wavelet transform was used to preprocess the one-dimensional signals. Finally, convolutional neural networks and attention mechanisms were used to analyze and recognize the features of multimodal data. In the article, a convolutional neural network based on an improved channel attention mechanism (ICAM-CNN) for one-dimensional signal fusion as well as a multimodal fusion network (MMFN) for multimodal data fusion was proposed, which could automatically detect node loss in the CZ silicon single-crystal growth process. The experimental results showed that the proposed methods effectively detected node-loss defects in the growth process of monocrystalline silicon with high accuracy, robustness, and real-time performance. The methods could provide effective technical support to improve efficiency and quality control in the CZ silicon single-crystal growth process.

## 1. Introduction

Monocrystalline silicon is an important semiconductor material that can be classified into solar-grade and IC-grade, based on its quality [1]. High-quality IC-grade monocrystalline silicon is the main material used for manufacturing integrated circuit chips. The Czochralski method is the main method used to prepare single-crystal silicon. With the development of ultra-large-scale integrated circuits, higher requirements have been placed on the quality and yield of silicon wafers, which need to increase in size while meeting industry requirements for resistivity, uniformity, crystal integrity, crystal orientation, oxygen, carbon content, purity, etc. Solar-grade monocrystalline silicon has lower quality requirements, but needs to be grown without dislocations. Dislocation defects are the core defects in the quality of monocrystalline silicon. Once dislocations occur during crystal growth, other quality parameters do not need to be considered [2]. Possible reasons for dislocation generation include impurities, fluctuations of unreasonable process parameters, etc. [3,4]. Node loss is the main manifestation of dislocations in the growth process of monocrystalline silicon and is also one of the basic judgment factors for the quality of CZ silicon ingots. Different crystal directions of single-crystal silicon have different numbers of nodes. Ingots grown in the [1 0 0] direction have four nodes. Figure 1 shows a normally grown ingot and two states of an ingot, one with a normal node and one with node loss. As Figure 1 demonstrates, a normally grown CZ silicon ingot has sharp nodes. However, the node of the node-loss crystal ingot disappears and the surface is very smooth.

During the growing process of a CZ silicon single crystal, the ingot needs to be regrown by re-melting and re-pulling if node loss occurs. If the loss can be promptly, quickly, and accurately identified, it not only saves costs, but also increases production efficiency. Otherwise, dislocation extension occurs and the ingot becomes polycrystalline, which results in the meaningless growth of crystals and significantly consumes production costs. Therefore, it is necessary and meaningful to research the node loss of CZ silicon single crystals.

Currently, most industrial sites still rely on manual inspections and identification to identify the node loss. Due to the high randomness of node loss, it is difficult to promptly and accurately detect node loss when relying on manual inspections. Certain companies only use related single-dimension information such as image features to identify node loss during the crystal growth process to achieve a higher detection efficiency, but the accuracy of this method is not high. Based on the conventional Faster R-CNN, Zhang et al. [5] used the ResNet-50 residual neural network as the backbone to extract the image features at the meniscus, which effectively improved the accuracy of the node—loss detection. However, their methods do not take into account the influence of directly related variables such as diameter, temperature, and pulling speed. Therefore, there is a large space for improvement in the node-loss detection field of CZ Silicon single crystals.

In recent years, with the rapid development of convolutional neural networks and multimodal data fusion technology, excellent research and achievements have emerged. The recent popular large-scale model ChatGPT is a typical representative of those achievements. Multimodal fusion technology in deep learning refers to the process of handling different forms of data where the model completes analyses and recognition tasks [6]. Its purpose is to establish a model that can handle and correlate information from multiple modalities, enabling machine-learning algorithms to comprehensively and efficiently understand the controlled object. Finally, the complementarity of diverse heterogeneous information can be realized and the limitations of single-modal data processing can be avoided. With the rise of the attention mechanism [7], the correlation between multimodal data has been extensively mined, enabling multi-models to fuse with higher accuracy, further improving the robustness and anti-interference ability of the model. Finally, the accuracy of recognition results is improved. Thanks to the technology of multimodal fusion, we proposed the method used in this paper on the basis of fully considering the correlation between the node-loss data of Czochralski silicon single crystal.

Due to the complexity of the silicon single-crystal growth process, it specifically includes the five stages of seeding, necking, shouldering, body, and tailing [8]. A large number of sensors were assembled to monitor the parameters of the silicon single-crystal growth process as well as the environmental parameters of a single-crystal furnace, thereby generating a large amount of historical data. In this paper, the ICAM-CNN method was used for the node-loss recognition of the one-dimensional diameter, temperature, and pulling speed signals directly related to node loss. The ResNet network was used for the node-loss recognition of the two-dimensional meniscus image. To further improve the accuracy, we take the advantages of one-dimensional signal feature extraction and two-dimensional image feature extraction to propose an MMFN method for multimodal data in the body process of a CZ silicon single crystal. Ultimately, the accuracy of node-loss detection could be improved and the timepoint of node loss could be identified in time, thereby saving production costs and improving production efficiency.

The main contributions of this paper are as follows:The required data were collected and preprocessed. The diameter, temperature, pulling speed signals, and image information at the meniscus directly related to the node loss of the silicon single crystal were measured using a variety of sensors. The continuous wavelet transform was also used to preprocess the signals of the diameter, temperature, and pulling speed.A convolutional neural network (ICAM-CNN) based on an improved channel attention mechanism was proposed. This method could be used to perform a feature fusion for the one-dimensional diameter, pulling speed, and temperature signals, finally determining the node-loss time.A two-dimensional image classification decision-making method based on the ResNet network was adopted. For the collected two-dimensional image information of the meniscus, the ResNet network was used as the image feature extraction network to extract deep image feature information to judge whether the system demonstrated node loss.A decision network based on multimodal data fusion—a multimodal fusion network (MMFN)—was proposed. MMFN first obtained the fusion features of the one-dimensional diameter, temperature, and pulling speed signals through ICAM-CNN, then obtained the two-dimensional image features through the ResNet network. In the feature fusion layer, MMFN used the concatenate method to achieve multimodal data feature-level fusion. Finally, a classifier was used to identify the node-loss time.A comparative discussion on the results of using single-modal and multimodal data fusion decisions was conducted. The results showed that using multimodal data fusion was more effective than any current single-modal data decision-making method. It could significantly improve the accuracy of CZ silicon single-crystal node-loss detection and meet the real-time and high-accuracy requirements of production sites.

## 2. Related Works

This article focused on the application of multimodal data fusion in the detection of the node loss of CZ silicon single crystals, aiming to overcome the problem of the low diagnostic accuracy of single-dimensional data.

At present, there are only a few articles researching the node loss of CZ silicon single crystals. The problem of the node loss of silicon single crystals is mainly studied and predicted from the aspects of the mechanism and the means of preventing node loss. Zhijun et al. [9] studied the problem of shoulders and broken edges in the preparation of 4-inch 111-oriented silicon single crystals and discussed the causes of dislocations. Dong et al. [10] analyzed dislocation formation from a theoretical point of view, discussing the specific reasons for node loss and bud drop. Choudhary et al. [11] studied the behavior of linear propagation dislocation along the growth direction in CZ silicon single crystals and also discussed the dislocation formation mechanism in heavily and lightly doped growth processes. Jing et al. [12] used a finite element numerical simulation to explain the cause of the liquid flow line in the front part of the shoulder and proposed a method of crystal transformation for the crystal-pulling process to reduce the occurrence of node loss. Du [13] used the method of data mining to establish an online prediction model for the problem of node loss in the body stage of silicon single crystals. Zhai et al. [14] proposed a feature-selection-based prediction study on node loss at the shouldering stage, but the data used did not directly correlate with the factors causing node loss, and the prediction accuracy was low.

The current approach to the problem of node-loss detection is mainly to extract features from the data, then perform classification and recognition tasks. The data feature extraction is mainly divided into one-dimensional data feature extraction and two-dimensional data feature extraction.

One-dimensional signal feature extraction methods are mainly divided into statistical feature-based, model-based, transformation-based, and fractal-based methods. Kankar et al. [15] extracted features such as statistical parameters and spectral features from vibration signals collected from faulty and healthy rolling bearings as inputs for different machine-learning algorithms. The performance of the model was evaluated according to indicators such as accuracy, sensitivity, and specificity, which had a certain contribution to the fault diagnosis using machine-learning techniques. In [16], the EEMD algorithm was used to decompose a signal into multiple intrinsic mode functions; these functions were then used as the input for a convolutional deep belief network. The method recognized the characteristics of different fault states by training the convolutional deep belief network and finally realized the automatic diagnosis of reciprocating compressor faults. Yang et al. [17] introduced the principle and algorithm of wavelet threshold denoising in detail, using an improved wavelet threshold denoising algorithm to suppress the noise in the signal and, at the same time, used the Savitzky–Golay filter to smooth the signal, resulting in a higher signal-to-noise ratio. Compared with other commonly used denoising methods, this method was more effective and robust.

For the feature extraction of two-dimensional images and with the rapid development of image processing technology, many scholars have focused on applying automatic inspection technology based on machine vision to inspection tasks. Traditional image processing algorithms have been proposed earlier or developed maturely, such as the principal component analysis (PCA) [18]. With continuous developments in the manufacturing industry, traditional algorithms have difficulty meeting the increasing detection accuracy and real-time requirements. In recent years, deep learning has achieved continuous progress in life as well as academic research. Deep learning models have shown strong feature extraction capabilities, gradually replacing traditional algorithms and becoming a research hotspot in the field of target detection and recognition. The most representative deep learning models are convolutional neural networks (CNNs) [19], recurrent neural networks (RNNs) [20], and autoencoders (AEs) [21], which can be applied to image classification tasks. In this paper, the convolutional neural network was mainly used to extract the features of the image and finally classify the image. With the popularity of CNNs, many evolutionary networks have been proposed. Examples include AlexNet, proposed in 2012 and published in 2017 [22]; VGGNet, proposed in 2014 [23]; and ResNet, proposed in 2015 and published in 2016 [24]. They all have excellent image feature extraction abilities.

For complex industrial field classification and recognition tasks, one-dimensional feature extraction and classification methods based on transformation theory and two-dimensional feature extraction and classification methods based on convolutional neural networks have shortcomings and cannot fully use the features of all data, resulting in poor model generalization ability. Multimodal data fusion technology can overcome this drawback using the complementary advantages of multimodal data and combining the features of one-dimensional signals and two-dimensional images for fusion decision-making. This can improve the generalization ability of the model, thus enhancing the detection accuracy.

Data fusion can be classified into three categories based on the level of information fusion, namely, data-level fusion, feature-level fusion, and decision-level fusion [25]. Feature-level fusion extracts features from signals collected from multiple sensors, then uses a network to analyze the feature information. After this, it forms a comprehensive feature set for use in the final target recognition and classification stage [26]. Compared with data-level and decision-level fusion, feature-level fusion is more flexible and can be combined with network structures, making it more versatile. In recent years, with the popularity of transformer technology [27], many attention mechanisms have been applied to fusion operations. Compared with traditional feature fusion methods, attention mechanisms can assign different weights to different parts of the input data, thereby using useful information in the data more effectively. It can automatically learn the relationship between features during the training process, thereby avoiding the tedious process of manually designing feature fusion rules. It can also adapt to different input dimensions and data types, making it highly applicable. References [28,29,30,31] all used attention mechanisms to fuse different features. By introducing attention mechanisms in the feature extraction process, the network can focus on important features with greater accuracy, thereby improving the effectiveness of features and the accuracy of classification and diagnosis. These studies all demonstrate that attention is an effective feature fusion method that can be used for classification tasks in various fields. To the best of our knowledge, no research has applied the attention module to node-loss detection in the growth process of single-crystal silicon.

## 3. Proposed Method

This article conducted a detailed study on the node-loss problem of a CZ silicon single crystal and collected, and preprocessed the node-loss data, and made a new dataset. Two new deep learning-based networks, ICAM-CNN and MMFN, were then proposed. The architecture of the proposed networks is described in detail below.

### 3.1. Data Collection and Preprocessing

#### 3.1.1. Data Collection

In the process of crystal growth, the node loss of a Czochralski silicon single crystal is random. There are many factors that lead to node loss, such as changes in thermal stress at the meniscus, the pulling speed, and temperature. As there is no technology at present to measure the thermal stress at the meniscus, we selected three measurable and directly related variables—temperature, pulling speed, and diameter—as the data sources to identify node loss.

The data in this article were all collected from crystal-growing equipment on site. The model of the single-crystal furnace was TDR-120CZ, as shown in Figure 2. The equipment could produce 100–310 mm CZ silicon single crystals. The diameter detection range was 4–350 mm. The maximum power of the equipment was 180 kw. The adjustment range of the crucible rotation speed was 0–15 rpm, the adjustment range of the crystal rotation speed was 0–20 rpm, the adjustment range of the crucible lifting speed was 0–0.5 mm/min, the adjustment range of the crystal lifting speed was 0–6 mm/min, the ultimate vacuum was 0.4 Pa, the adjustment range of the air intake was 0–150 L/min, and the adjustment range of the liquid level was 0–100 mm. The lifting stroke of the crystal was 2.8 m and the lifting stroke of the crucible was 600 mm. As far as the process parameters on-site were concerned, the target setting value of the diameter was 308 mm, the adjustment range of the casting speed was 0.2–1.1 mm/min, and the temperature adjustment was determined according to the input of the amount of silicon material. The general rule was to cool down first, then raise the temperature. The adjustment range was usually 2100–2500 (dimensionless).

The crystal diameter signal used for the data fusion was acquired using a high-temperature infrared pyrometer sensor. The crystal growth temperature signal was acquired using a RAYTEK Marathon FR Infrared Thermometer sensor. The pulling speed of the crystal was obtained after conversion using Schneider XCC Series Absolute Encoders. The data of the temperature, pulling speed, and diameter were sampled every 2 s. The image data during the growth process were obtained using a Microvision MV-300UC camera. It had a resolution of 300 megapixels and used a CMOS imaging method with a frame rate of 15 fps. All acquisition devices are shown in Figure 2.

In the growth process, once the temperature and pulling speed exhibit large mutations, it leads to a greater risk of node loss. There is also an immediate reaction in the diameter. As shown in Figure 3, the horizontal axis was the sampling time from the isodiametric process and the vertical axis was the diameter signal. Node loss occurred at the crystal body stage. It was not difficult to ascertain from the diameter signal that once there was node loss, the frequency of the crystal diameter change would be altered.

For normal on-site production, inspectors usually judge node loss according to the images. If there is node loss, the changes can be observed from the image of the meniscus captured by the camera. Figure 4 is the image data of the crystal body stage obtained using a TDR-120CZ single-crystal furnace. The image on the left is the meniscus taken during normal growth, with periodic bump information. The image on the right is the meniscus was taken when the node was lost, which was very smooth without any bump features.

#### 3.1.2. Data Preprocessing

Through multiple crystal-pulling experiments, 1800 sets of temperature, pulling speed, and diameter near the time of node loss of different lots as well as images of the meniscus of the crystal at the corresponding time were collected. When the crystal grew normally, 1844 datasets of temperature, pulling speed, and diameter signals as well as the image data of the meniscus at the corresponding time were collected. Due to the characteristics of a long delay and a large lag in the crystal growth process, the temperature, pulling speed, and diameter signals in each set of data were represented by the data segment from the previous 7 min, that is, 210 data points.

To highlight the time–frequency changes of the diameter, temperature, and pulling speed signals and more accurately identify node loss, the continuous wavelet transform was used and the data preprocessing method shown in Figure 5 was proposed.

First, we took the difference of each input dataset to obtain the change in temperature, diameter, and pulling speed every 2 s. Then, the continuous wavelet transform was used in the signal processing to fully extract the time–frequency domain characteristic changes of the temperature, pulling speed, and diameter signals. Finally, a processed time–frequency spectrum of the temperature, diameter, and pulling speed signals was obtained.

The formula for continuous wavelet transform is as follows:(1)$$WT_{a,b}(t)=\int_{-\infty }^{+\infty }f(t)•\varphi_{a,b}{}^{*}(t)dt$$
where f(t) is the original function, φ(t) is the wavelet basis function, a is the scale parameter, and b is a time parameter. The specific form of the function is as follows:(2)$$\varphi_{a,b}(t)=\frac{1}{\sqrt{a}}\varphi (\frac{t-b}{a})$$
where a is the scale parameter and b is a time parameter. Through the transformation, the time subdivision of the signal at a high frequency and the frequency subdivision at a low frequency is finally achieved, which can adapt to the requirements of the time–frequency signal analysis and focus on any detailed characteristics of the collected signals.

### 3.2. Method 1: ICAM-CNN

The one-dimensional signals included the diameter, temperature, and pulling speed. For one-dimensional signal fusion node-loss detection, a network structure based on ICAM-CNN was proposed, as shown in Figure 6.

First, the data were preprocessed by the continuous wavelet transform. Then, the time–frequency spectrogram was inputted into the CNN for training. The CNN was mainly composed of convolutional layers, pooling layers, fully connected layers, and decision layers. The features of the data were extracted by convolution and pooling operations. Below is the main calculation formula for the convolution operation.
(3)$$Z_{i}^{k}=f(W_{i}^{k}⊗Z_{i-1}^{k}+b_{i}^{k})$$

Here, $Z_{i}^{k}$ represents the feature map formed by the convolution kernel of the ith layer, $W_{i}^{k}$ denotes the weight matrix, $b_{i}^{k}$ represents the bias, ⊗ denotes the convolution operation, and f represents the ReLU activation function.

Then, the improved channel attention mechanism, which is shown in Figure 7, was used to perform a correlation fusion analysis on the diameter, temperature, and pulling speed features extracted by the convolutional neural network. The calculation formula for the improved channel attention is as follows:(4)$$\begin{array}{ll} M_{c} & =f(MLP(Map_{avg})+MLP(Map_{\max}))\\ & =f(V_{avg}+V_{\max}) \end{array}$$
where f represents the SoftMax function, $Map_{avg}$ and $Map_{\max}$ represents the global average pooling feature and the global maximum pooling feature, $V_{avg}\in R^{c\times 1\times 1}$ and $V_{\max}\in R^{c\times 1\times 1}$ represents one-dimensional vectors further processed by the *MLP*, $M_{c}\in R^{c\times 1\times 1}$ represents a final evaluation score for each channel.

Finally, we sent the correlation fusion features processed by the ICAM-CNN network into the classifier for decision-making, and obtained the result of using the one-dimensional diameter, temperature, and pulling speed signals for the node-loss detection.

### 3.3. Method 2: MMFN

On the basis of one-dimensional node-loss fusion detection, two-dimensional meniscus image data were introduced and a fusion decision method based on multimodal data features was proposed to further improve the accuracy of node-loss detection. The specific network structure is shown in Figure 8.

First, using the ICAM-CNN method, the features for all the one-dimensional signals were extracted. Then, using the ResNet network, the features for all the two-dimensional meniscus images were extracted. After, feature concatenation technology was used to fuse the extracted features at the feature layer, achieving a feature-level fusion of the multimodal data. Finally, a fusion decision based on the features of the multimodal data was achieved through a classifier.

For the two-dimensional image data, the ResNet network, which is shown in Figure 8, was used as the feature extraction network. It is mainly aimed at solving the problems of gradient disappearance and model degradation in deep networks. The residual learning structure consists of a feedforward neural network and a skip connection method, as shown in Figure 9.

The forward propagation formula for the residual structure is:(5)H(x)=F(x)+x
where, x represents the input, F(x) represents the residual, and H(x) represents the desired target. Through such a residual structure, the number of layers can be continuously superimposed to improve the accuracy of the final node-loss detection.

## 4. Experimental Setup and Result

### 4.1. Model Training

The validity of the method was verified using the Czochralski silicon single-crystal node-loss dataset collected and produced by ourselves. The algorithms proposed in this paper were implemented using the open-source TensorFlow deep learning framework. The CPU used was Intel(R) Xeon(R) Gold 5318Y CPU @ 2.10 GHz (2 processors) and the GPU used Nvidia Grid V100D-8Q.

A total of 3644 datasets of temperature, casting speed, diameter, and images were collected in the one-dimensional dataset, including 1800 sets of node-loss data and 1844 sets of normal growth data. Each dataset of pulling speed, temperature, diameter, and images shared the same label. The split ratio of the training set, validation set, and testing set in the data was 7:2:1 and the size of the input image was normalized to 224 × 224. The Adam, proposed by Kingma et al. [32], was used as the overall optimizer and the backpropagation algorithm to realize the optimization of the entire network model. For training, the cross-entropy loss function was selected as the loss function. After parameter optimization, the batch size of all datasets was set to 32 and the epoch was set to 20. To avoid overfitting, the dropout was set to 0.5 in the FC layer of the proposed network model and the L2 regularization coefficient of the convolution kernel was set to 0.01 in the convolution layer. The data enhancement on the collected image information of the meniscus was also performed.

### 4.2. Evaluation Method

This article drew a confusion matrix on the final test set and calculated the recall rate (Recall), precision (Precision), F1 score, and accuracy (Accuracy) as evaluation indicators to accurately evaluate the effectiveness of the proposed method.

The calculation formulas for these values are as follows.
(6)$$Recall=\frac{TP}{TP+FN}$$
(7)$$\mathrm{Precision}=\frac{TP}{TP+FP}$$
(8)$$F1Score=\frac{P\times R}{2(P+R)}$$
(9)$$Accuracy=\frac{TP+TN}{TP+TN+FP+FN}$$

In the above formulas, *TP* represents the cases where the classifier correctly identified positive samples as positive, *TN* represents the cases where the classifier correctly identified negative samples as negative, *FP* represents the cases where the classifier incorrectly identified negative samples as positive, and *FN* represents the cases where the classifier incorrectly identified positive samples as negative.

### 4.3. Analysis of the Training Results

#### 4.3.1. The Result of CNN and ResNet

An ordinary CNN network was used to separately formulate decisions on the diameter, pulling speed, and temperature signals. We then obtained the decision result of a single signal. After, the ResNet network was used to obtain the results of the two-dimensional image data. Among the 3644 sets of data, there were 730 sets in the testing set.

The structure diagram of different inputs is shown in Figure 10.

The historical curves of the training set and the validation set are shown on the left of Figure 11 and the confusion matrixes of the testing sets are shown on the right of Figure 11. The calculated evaluation indicators are shown in Table 1.

As seen in Figure 11a–d and Table 1, the model started to converge after 20 rounds of training. The highest accuracy rate of 95.48% was obtained by relying on the diameter signal. Combined with the F1 Score, the overall recognition rate and the classification effect were the best depending on the diameter signal. A reliance on pulling speed followed. Compared with the pulling speed signal, the diameter signal did not have a hysteresis effect on node loss; therefore, the recognition accuracy was higher, which coincided with the theory and practice. Due to the large hysteresis effect of temperature on node loss and as the crystal length continued to grow, the ability of the system to resist temperature disturbances continued to increase. Therefore, relying on a temperature classification achieved poor results, with an accuracy rate of only 74.11%.

The effect of training with the ResNet network on the two-dimensional image data was not ideal, with the lowest accuracy of only 67.12%. The model quickly converged, and the training accuracy curve significantly fluctuated. From the Recall value, the recognition rate of the abnormal data was only 33.89% and the false detection rate was high. Compared with relying on manual inspections and identification on an industrial site, the effect was improved, but not by much. Objectively speaking, the low classification accuracy was mainly due to unclear image features.

#### 4.3.2. The Result of Method 1: ICAM-CNN

To verify the effectiveness of the proposed ICAM-CNN method, it was compared with the concatenate method to directly splice the feature layers of the diameter, pulling speed, and temperature signals.

The structure diagram of the different algorithms is shown in Figure 12.

The historical curves of the training set and the validation set are shown on the left of Figure 13 and the confusion matrixes on the testing set are shown on the right of Figure 13. The calculated evaluation indicators are shown in Table 2.

From Figure 13 and Table 2, it could be observed that using the diameter, pulling speed, and temperature for multi-sensor mixed training was more effective and had a higher accuracy compared with using single diameter, temperature, and pulling speed data separately to formulate a node-loss decision. Table 2 shows that the accuracy of the direct fusion of multiple signals using the concatenate method was not as high as that of the ICAM-CNN network using ICAM for correlation fusion. The network with the attention mechanism had a fast convergence speed and could perform feature extraction and correlation fusion on different sensor data more effectively.

#### 4.3.3. The Result of Method 2: MMFN

To further improve the accuracy of classification recognition, enhance the generalization ability of the model, and fully use the features of multiple types of data, we fused the features obtained by the ResNet and ICAM-CNN networks at the final feature layer, based on one-dimensional signal node-loss detection and two-dimensional image node-loss detection. Then, the final multimodal data fusion results using MMFN were obtained.

The structure diagram of MMFN is shown in Figure 14.

Figure 15 shows the training results using this approach. The calculated evaluation indicators are shown in Table 3.

Based on the overall experimental results, an accuracy of 97.95% was achieved by using the ICAM-CNN network for single-modal fusion. By introducing image data features, an accuracy of 98.36% was achieved. Looking at the F1 Score, MMFN achieved a result of 98.31% on the testing set, which was more successful than the result obtained by ICAM-CNN and more successful than any result obtained using a single-dimensional signal.

## 5. Conclusions

Using machine-learning methods to identify CZ silicon single-crystal node-loss detection is a novel and challenging task. In this paper, the continuous wavelet transform was used to preprocess one-dimensional diameters, pulling speeds, and temperature signals during the crystal growth process. A one-dimensional signal fusion decision-making method ICAM-CNN was proposed, which achieved an accuracy rate of 97.75%. Compared with the accuracy rate of 95.48% obtained by the CNN for pure diameter signal recognition, the accuracy rate of 81.23% for pure pulling speed signal recognition, and the 74.11% accuracy rate for pure temperature signal recognition, this method achieved more accurate results. On this basis, the image information captured by the two-dimensional image sensor was introduced and the one-dimensional signals were combined with the two-dimensional image data for mixed training to achieve multimodal data fusion decisions. On our self-made dataset, the MMFN finally achieved an accuracy of 98.36%. Compared to the 67.12% accuracy achieved by using Resnet network recognition in pure two-dimensional meniscus images and the 97.95% accuracy achieved by using ICAM-CNN in one-dimensional signals fusion, it has improved and can verify the effectiveness of the method. Compared with the manual inspection at the production site, the method used in this paper is more accurate and has higher real-time performance, which can meet the real-time and accurate requirements in field production. The method is of great significance for improving the automation of single crystal furnaces, reducing the labor intensity of manual inspection, and preventing production accidents, and has practical industrial value.

Since the data used in this paper are all collected in the industrial field, there is no need to arrange additional sensors, which have the conditions for implementation in the industrial field. In the future, based on the existing research, we plan to further enrich the node-loss dataset according to the field operation results, and constantly update the node-loss detection method iteratively to improve the accuracy, generalization and robustness of the model. At the same time, we plan to introduce more variables related to the node-loss to adapt to the crystal growth process with variable crucible rotation and even variable magnetic field strength. Finally, on the basis of the node-loss detection, we will consider the combination of data and mechanism to carry out the research on node-loss prediction.

## Figures and Tables

**Figure 1 sensors-23-05855-f001:**
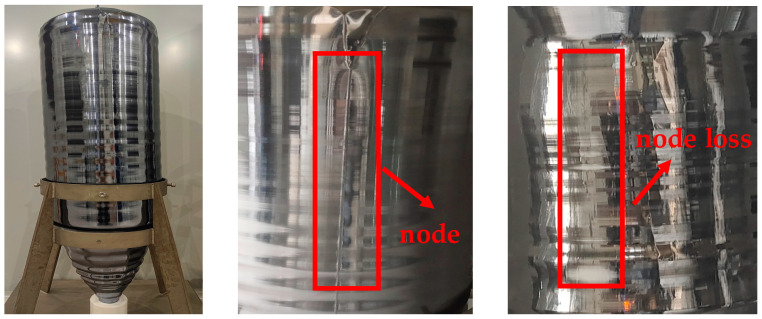
Whole normal ingot (**left**), normal node (**middle**), and node loss (**right**).

**Figure 2 sensors-23-05855-f002:**
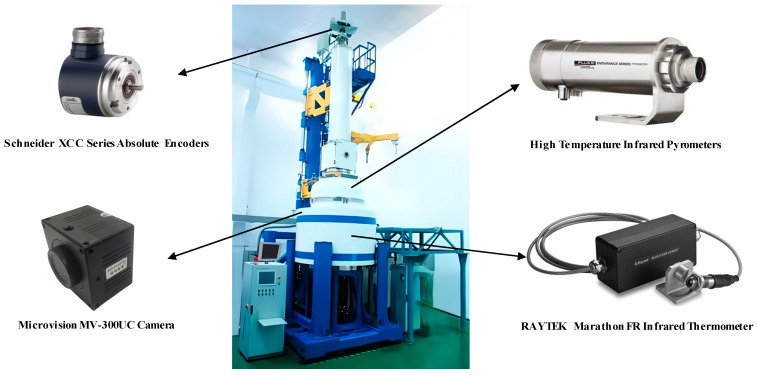
Capture device images.

**Figure 3 sensors-23-05855-f003:**
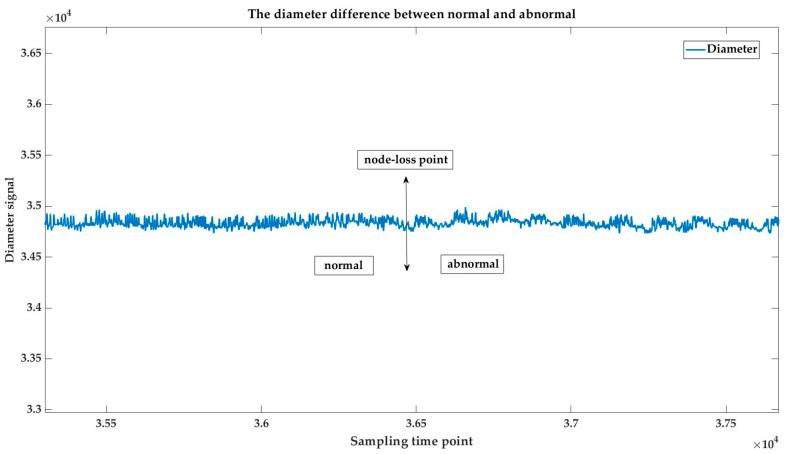
Diameter difference between normal and abnormal.

**Figure 4 sensors-23-05855-f004:**
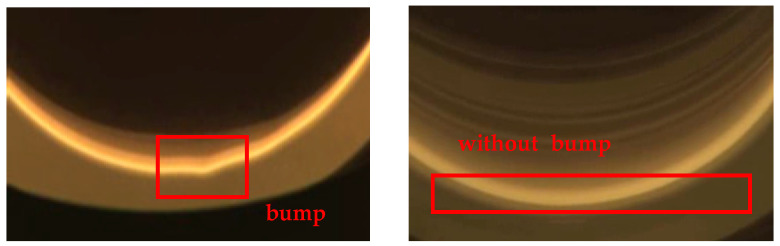
The difference at the meniscus when normal (**left**) and with node loss (**right**).

**Figure 5 sensors-23-05855-f005:**
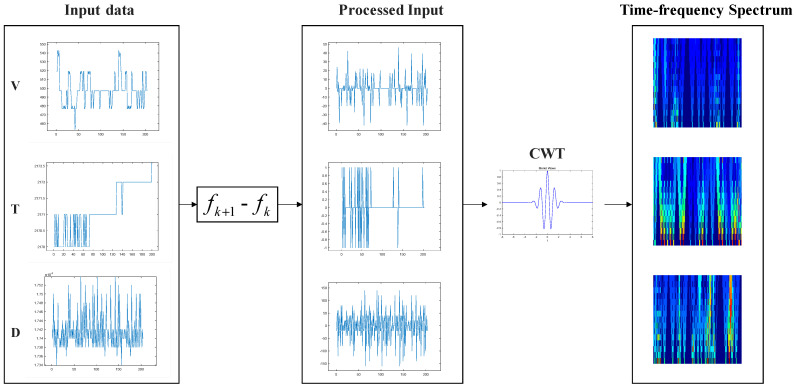
Continuous wavelet transforms for one-dimensional signal preprocessing.

**Figure 6 sensors-23-05855-f006:**
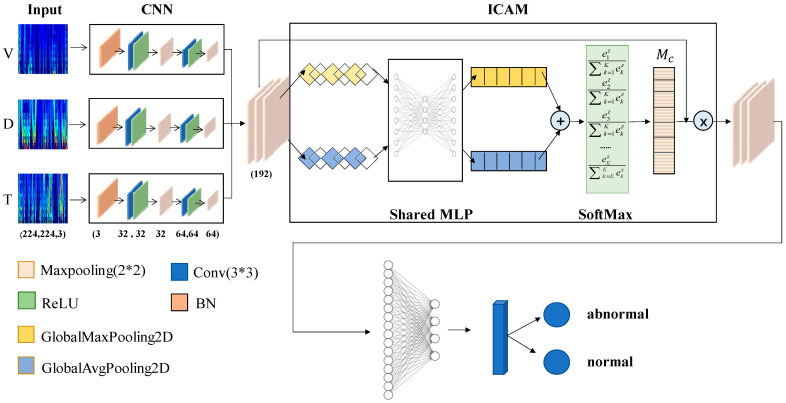
ICAM-CNN network structure.

**Figure 7 sensors-23-05855-f007:**
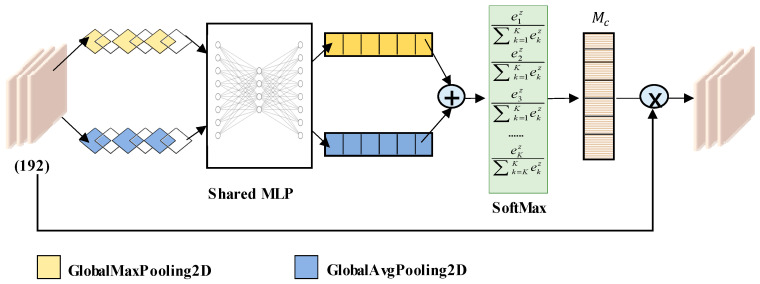
ICAM structure.

**Figure 8 sensors-23-05855-f008:**
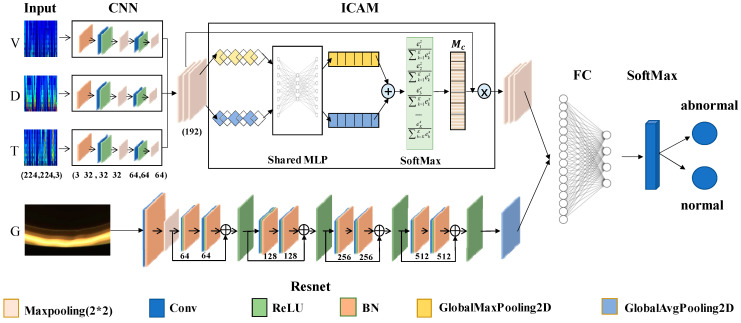
MMFN structure.

**Figure 9 sensors-23-05855-f009:**
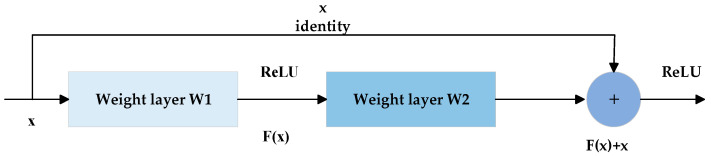
Residual block structure.

**Figure 10 sensors-23-05855-f010:**
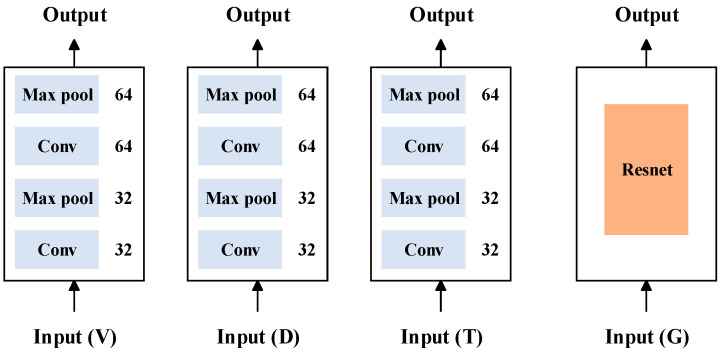
Structure diagram of different inputs.

**Figure 11 sensors-23-05855-f011:**
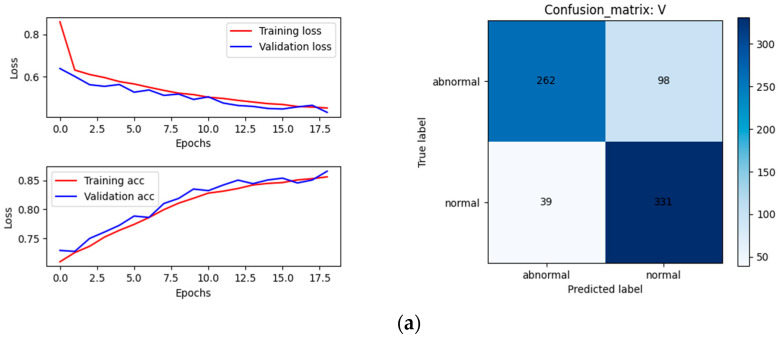
Results of separate single-modal data: (**a**) ordinary CNN results of pulling speed V; (**b**) ordinary CNN results of temperature T; (**c**) ordinary CNN results of diameter D; (**d**) ResNet results of meniscus image G.

**Figure 12 sensors-23-05855-f012:**
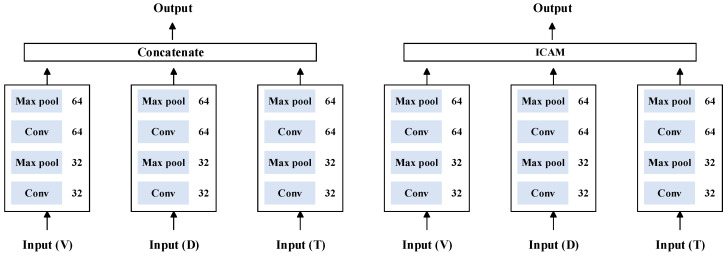
Structure diagram of different algorithms.

**Figure 13 sensors-23-05855-f013:**
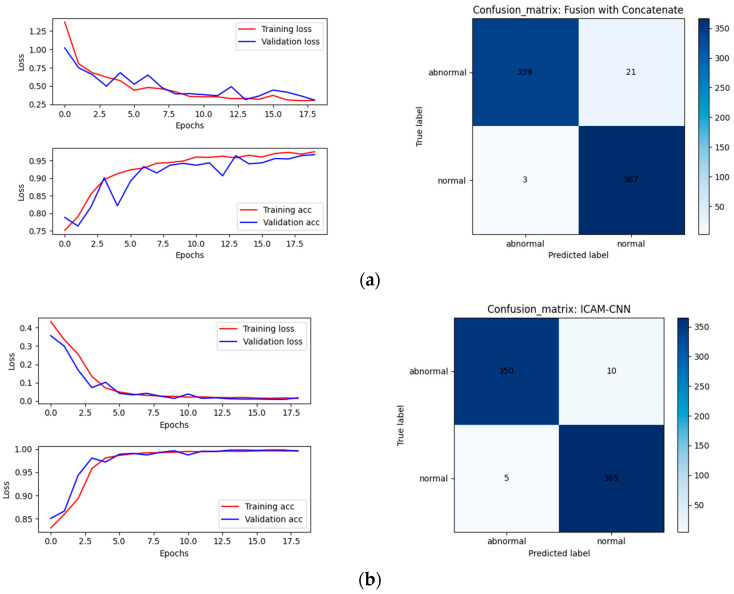
Results of one-dimensional signals using different fusion methods: (**a**) results of one-dimensional signals using concatenate fusion method; (**b**) results of one-dimensional signal ICAM fusion method.

**Figure 14 sensors-23-05855-f014:**
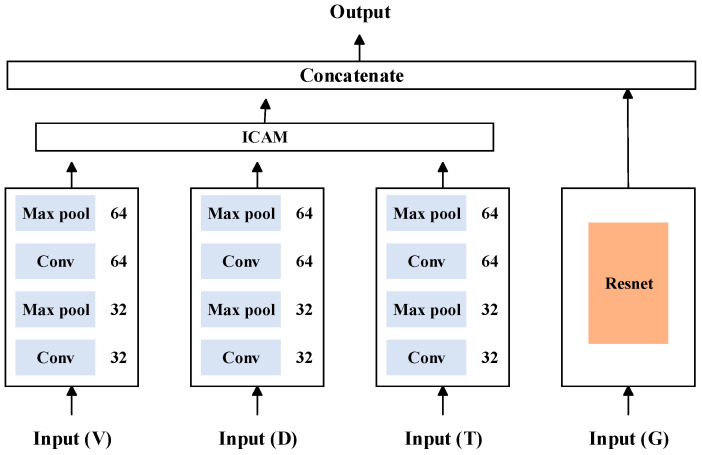
Structure diagram of MMFN.

**Figure 15 sensors-23-05855-f015:**
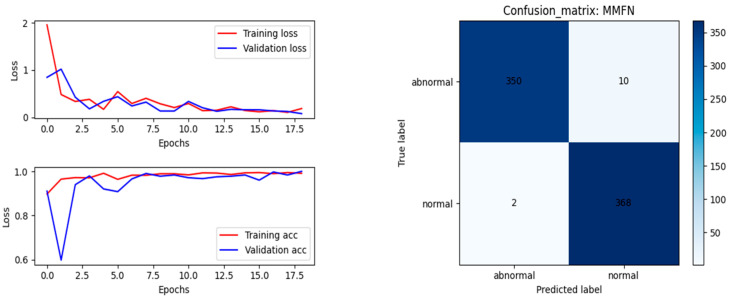
Results of the multimodal data fusion network.

**Table 1 sensors-23-05855-t001:** Classification results for single-modal data without any fusion method.

		Accuracy	Recall	Precision	F1 Score
V	Abnormal	81.23%	72.78%	87.04%	79.27%
	Normal		89.46%	77.16%	82.85%
T	Abnormal	74.11%	70.56%	75.37%	72.89%
	Normal		77.57%	73.03%	75.23%
D	Abnormal	95.48%	94.44%	96.32%	95.37%
	Normal		96.49%	94.70%	95.58%
G	Abnormal	67.12%	33.89%	98.39%	50.41%
	Normal		99.46%	60.73%	75.41%

**Table 2 sensors-23-05855-t002:** Evaluation indicators using different fusion methods.

		Accuracy	Recall	Precision	F1 Score
Concatenate	Abnormal	96.71%	94.17%	99.12%	96.58%
	Normal		99.19%	94.59%	96.83%
ICAM	Abnormal	97.95%	97.22%	98.59%	97.90%
	Normal		98.65%	97.33%	97.99%

**Table 3 sensors-23-05855-t003:** Evaluation indicators using a multimodal data fusion network.

		Accuracy	Recall	Precision	F1 Score
MMFN	Abnormal	98.36%	97.22%	99.43%	98.31%
	Normal		99.46%	97.35%	98.40%

## Data Availability

The data presented in this study are available on request from the corresponding author. The data are not publicly available due to legal considerations.

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
