# Peer review of "Node-Loss Detection Methods for CZ Silicon Single Crystal Based on Multimodal Data Fusion"

_sensors, 2023, doi:10.3390/s23135855_

Round 1
Reviewer 1 Report
The paper deals with the Research on Node Loss Detection Method of CZ Silicon Single
Crystal Based on Multimodal Data Fusion.
According to the reviewer, the paper is worth publishing at Sensors Journal,
but corrections are needed and then the paper can be accepted for publication in the journal.
While the authors have made considerable research effort,
the presentation of the paper must be improved.
Additionally make the following corrections to the manuscript:
Comment 1
Line 10
In the CZ (Czochralski) method
The authors should replace
In the Czochralski (CZ) method
Line 68
the controlled object, Finally, the complementarity
Extended text editing
the controlled object. Finally, the complementarity
Extended text editing:
The authors must format the paper according to the journal's instructions
Line 45
Figure1 The whole normal ingot(left), node normal (middle) , node loss (right)
The authors must replace
Figure 1. The whole normal ingot (left), node normal (middle), node loss (right)
Line 213
Figure2 Diameter difference between normal and node loss
The authors must replace
Figure 2. Diameter difference between normal and node loss
The same comment for the rest of the Figures and the Tables.
Figures 11, 13 and 15
Input(V).......
The authors must replace
Input (V).......
Lines 99 - 100
work(MMFN),
The authors must replace (insert a space)
work (MMFN),
Comment 2
Lines 16, 18, 75, 83, 104, 200, 201, 240, 243, 249, 255, 256, 258, 296, 311, 316, 317, 318, 339, 339, 360, 373, 374, 417, 431, 444, 454, 456, 463, 466, 469, 475 and 477
It is not so good to use the word "we".
The authors must rephrase.
Comment 3
The authors must comment the ref. [5] (is missing).
Comment 4
Line 117, 119, 121, 124, 128, 138 and 147
There is not a author at the Ref. [8], [9], [10], [11], [13], [14] and [16]
The authors must insert the "et al.", for examble
Pei [8]
The authors must replace
Pei et al. [8]
Comment 5
The authors must check:
Lines 166 - 167
such as Alexnet proposed in 2012 [21], (or 2017?)
Line 167
Resnet proposed in 2015[23].
The authors must check:
Resnet proposed in 2016 [23].
Comment 6
The authors must give more details how the Figure 2 occur, and they must insert the axis x and axis y, the units and explain the 104.
The authors must give more details how the Figure 3 occur (experiment with the using equipment).
Comment 7
Line 224
The authors must give more details for the limits of the diameter, pulling speed, temperature signal and the meniscus image data.
Line 228
The authors must give more details for the equipment (type, model)
Comment 8
The authors must give a typical result for the experiment.
Also, the authors must give the whole results for all the data in the Supplementary Materials.
Major problem:
How the paper is presented, is like a closed box, without being able to check the reliability of the method proposed by the authors.
Comment 9
Line 329
The "Kaiming He et al. in 2015." is not at the References. The authors must insert the paper at the References.
Comment 10
Line 355
uses Adam as the overall
The authors must explain the word "Adam" with more details.
Comment 11
Line 357
the batch size of all datasets is set to 32, and the epoch is set to 20.
The authors must explain why they choose those values.
Comment 12
The authors must format the References according to the journal's instructions
References should be described as follows, depending on the type of work:
Journal Articles:
1. Author 1, A.B.; Author 2, C.D. Title of the article. Abbreviated Journal Name Year, Volume, page range.
The authors must delete the "[J]" and "C".
Reviewer 2 Report
The work titled ' Research on Node Loss Detection Method of CZ Silicon Single Crystal Based on Multimodal Data Fusion ' is related to the detection of loss of monocrystalline silicon nodes in industrial objects. Based on the analysis of diameter, temperature, pulling speed, and two-dimensional images of the bent sample surface, the authors have developed a methodology for determining the presence or absence of monocrystalline silicon node loss. The methodology is largely based on machine learning algorithms. Satisfactory agreement has been found between predictive data and experimental data. In my opinion, the work is interesting and brings new scientific elements to the optimization of the monocrystalline manufacturing process. No substantive errors have been identified. However, before publication, the authors should address the following comments:
1. On what basis was the number of training data determined? The authors should provide an answer to this question to explain the rationale for selecting the training data.
2. Was the risk of overfitting or noise robustness examined in the developed procedures? The authors should discuss whether research was conducted in these areas and what results were obtained.
Furthermore, the axis of the graph presented in Figure 2 is not described. The variables 'a' and 'b' in Equations 1 and 2 are not explained.
The authors should take these comments into account to ensure clarity and precision in the presentation of their work.
Reviewer 3 Report
First of all, an important topic on using machine learning methods to identify CZ silicon single crystal node loss detection. However, I would suggest reconsidering the paper after inviting the authors to respond to the following comments.
1. A lot of excessive information, which makes it difficult to read and follow. It is better to focus on the most important findings. The article would benefit from a more condensed style.
2. P363-364: The calculation formulas for the recall rate, precision, F1 score, and accuracy are different, there should be numbered one by one, rather than as a whole.
3. English must be improved through the article. I would suggest the authors ask a native for correct English, and then submit again thereafter.
4. References must be formatted according to journal style. Use the standard abbreviations for journal names given in the International Standard ISO 4.
For examples:
(1) Somewhere the authors list “four authors” e.g. the reference [18], somewhere “three authors with et al.” e.g. references [13],[25],[30]. Be consistent.
(2) The “and” should lie in the second author and the third, rather than between the first and second author in reference[24].
5. Introduction needs better orientation while writing with three steps- state of art process, research gaps and highlighting the current innovative step adopted.
6. Please indicate with arrows in Figure 1 what where what.
7. Please give the meaning of symbols in the formulas, particularly in equations (1) to (2).
8. Please give some specific and quantitated information in the conclusion.
English must be improved through the article, particularly some miswriting, grammatical and conciseness issues.
Round 2
Reviewer 1 Report
Comment 1
Lines 131 - 132
Pei et
al. [9] studied
The authors must replace
Zhijun et
al. [9] studied
Comment 2
Line 138
Zhang et al. [12] used a finite
The authors should replace
Jing et al. [12] used a finite
Comment 3
Line 353
The ResNet network was proposed by Kaiming He et al. in 2015.
The authors should replace
The ResNet network was proposed by Kaiming He et al. in 2015 [24].
Comment 4
Line 378
Adam [32] was used
The authors should replace
Kingma et al. [32] was used
Reviewer 3 Report
I think the authors have taken into consideration the comments and suggestions of the reviewers of the original manuscript and attempted to address them all. However, some new questions also have existed and the quality of this paper should be improved. The related suggestion is as follows:
1. Please double-check the equation of SoftMax in Figure 7 as it is different from that in Figure 8.
2. I also think too much detailed information in the manuscript makes it hard to follow at times. For example, the authors give much detailed information about Two deep learning-based networks, ICAM-CNN and MMFN in Section 3.2-3.3. please delete the excessive information and re-constructed them in a more condensed style.
3. Please make a comparison between the two networks of ICAM-CNN and MMFN based on Training Results and give information about the feasibility of further application of detecting node loss while adapting to the crystal growth process.
No.
